# Cross-Linking with Polyethylenimine Confers Better Functional Characteristics to an Immobilized β-glucosidase from *Exiguobacterium antarcticum* B7

**Ricardo Rodrigues de Melo [1,2], Robson Carlos Alnoch [1,3], Amanda Silva de Sousa [2], Hélia Harumi Sato [4], Roberto Ruller [5] and Cesar Mateo [1,*]**

1. Departamento de Biocatálisis, Instituto de Catálisis y Petroleoquímica (CSIC), Marie Curie 2. Cantoblanco, Campus UAM, 28049 Madri, Spain; ricardorodriguesmelo@gmail.com (R.R.d.M.); robsonalnoch@hotmail.com (R.C.A.)
2. Laboratório Nacional de Ciência e Tecnologia do Bioetanol (CTBE), Centro Nacional de Pesquisa em Energia e Materiais (CNPEM), CEP 13083-970 Campinas, São Paulo, Brazil; amanda.ssousa1@gmail.com
3. Departamento de Bioquímica e Biologia Molecular, Universidade Federal do Paraná, Cx. P. 19081 Centro Politécnico, CEP 81531-980 Curitiba, Paraná, Brazil
4. Departamento de Ciência de Alimentos, Faculdade de Engenharia de Alimentos, Universidade Estadual de Campinas (UNICAMP), CEP 13083-862 Campinas, São Paulo, Brazil; heliah@fea.unicamp.br
5. Instituto de Biociências, Universidade Federal de Mato Grosso do Sul (UFMS), CEP 79070-900 Campo Grande, Brazil; robertoruller@ufms.br
* Correspondence: ce.mateo@icp.csic.es; Tel.: +34-915854768; Fax: +34-915854860

**Abstract:** β-glucosidases are ubiquitous, well-characterized and biologically important enzymes with considerable uses in industrial sectors. Here, a tetrameric β-glucosidase from *Exiguobacterium antarcticum* B7 (*Ea*BglA) was immobilized on different activated agarose supports followed by post-immobilization with poly-functional macromolecules. The best result was obtained by the immobilization of *Ea*BglA on metal glutaraldehyde-activated agarose support following cross-linking with polyethylenimine. Interestingly, the immobilized *Ea*BglA was 46-fold more stable than its free form and showed optimum pH in the acidic region, with high catalytic activity in the pH range from 3 to 9, while the free *Ea*BglA showed catalytic activity in a narrow pH range (>80% at pH 6.0–8.0) and optimum pH at 7.0. *Ea*BglA had the optimum temperature changed from 30 °C to 50 °C with the immobilization step. The immobilized *Ea*BglA showed an expressive adaptation to pH and it was tolerant to ethanol and glucose, indicating suitable properties involving the saccharification process. Even after 9 cycles of reuse, the immobilized β-glucosidase retained about 100% of its initial activity, demonstrating great operational stability. Hence, the current study describes an efficient strategy to increase the functional characteristics of a tetrameric β-glucosidase for future use in the bioethanol production.

**Keywords:** β-glucosidase immobilization; tetrameric enzyme; glutaraldehyde; polyethylenimine; pH-stability; cellobiose hydrolysis

---

## 1. Introduction

Cellulose, the most abundant component of plant biomass on the Earth, is the primary product of photosynthesis and the most abundant renewable bio-resource produced in the biosphere [1]. Cellulosic biomass has a great industrial relevance, mainly due to its saccharified form, that is, glucose units useful for use by yeasts and other organisms to produce bioethanol or other biomaterials [2,3]. The most important enzymes involved in the biodeconstruction of cellulose fibres, cellulases, cleave the

β-1,4-glycosidic bonds of cellulose to glucose molecules. In a classical view, cellulases are catalogued according to their mechanisms of action as endoglucanases (EGLs, EC 3.2.1.4); cellobiohydrolases (CBH I, EC 3.2.1.176 and CBH II, EC 3.2.1.91); and lastly, β-glucosidases (BGLs, EC 3.2.1.21) [4,5]. BGLs are key hydrolases in the conversion process, since its reaction is considered to be the rate-limiting step in the enzymatic hydrolysis of cellulose polymers. Thus, the function of BGLs is not only to convert cellobiose to glucose but also to reduce the cellobiose inhibition, resulting in efficient functioning of other enzymes (EGLs and CBHs) [6].

BGLs are ubiquitous proteins found in all kingdoms of life [3]. β-glucosidases are catalysts involved in a variety of natural processes, being extensively explored in different biotechnological procedures in industrial level. Nowadays, β-glucosidases have been drawing intensive attention because of their critical role in the biological conversion of cellulose fibres to glucose [2]. Besides, the reactions with BGLs can be particularly useful in industrial processes in the pharmaceutical field (e.g., production of bioactive agents and antimicrobial compounds for use in cosmetics), as well as in food areas (e.g., processing of wines, teas or fruit juices and aroma enhancement) [2,7–10].

Recently, a novel GH1 β-glucosidase from the psychrophilic bacterium *Exiguobacterium antarcticum* B7 (namely, *Ea*BglA) with higher *kcat* over mesophilic counterparts at 30°C was described and functionally characterized [11]. *Ea*BglA showed a cold-active mechanism and also a high glucose tolerant capacity (up to 1 M of glucose, [11]), desirable features in bioprocesses with high glucose production, such as biomass saccharification [12]. Structural analyses revealed that *Ea*BglA has a tetrameric arrangement, being the first kind to be reported within the GH1 family [13]. Although *Ea*BglA has interesting features as biocatalyst, it is still necessary to conduct studies for its economically viable application.

A notorious alternative to develop a more cost-effective enzymatic process has been the use of immobilization methods [14,15]. The benefits of enzyme immobilization, such as longer shelf life, reusability and stability against temperature and pH variations have been extensively studied [16]. However, although the immobilization techniques have been applied as efficient tools [17–20], increases in demand for new enzymatic processes and the structural diversity found in the new biocatalysts, means that the materials and/or methodologies used in the immobilization also have to be updated constantly. Thus, it is of fundamental importance to identify new techniques that combine mechanical properties and biocompatibility that can improve enzyme characteristics.

Many studies on β-glucosidase immobilization have been reported. The most common immobilization strategies are adsorption or covalent attachment using different supports as amino-glutaraldehyde-agarose [21], eupergit C [22], glyoxyl-agarose [23], magnetic nanoparticles [24,25] and chitosan [26,27]. Moreover, studies of directed immobilization via biotinylation [28] and his-tagged region [29,30] have been described as an interesting tool. The specific quaternary arrangement demonstrated by *Ea*BglA from *E. antarcticum* B7 may be required to exhibit its desired function [13]. Thus, we consider that the immobilization step on glutaraldehyde supports may be a good option as a simple non-distorting immobilization method, since it occurs under very mild reaction conditions (neutral pH), which makes the technique generally very gentle producing catalysts with good recovered activities [31]. In the current work, immobilization on different activated agarose supports followed by post-immobilization with poly-functional macromolecules was applied to stabilize a tetrameric β-glucosidase from the *E. antarcticum* B7. To access the modifications on the structural and catalytic efficiency obtained with immobilization step, a study of thermal stability was performed. Subsequently, optimum pH, temperature, pH-tolerance, sugar and ethanol tolerance, storage and conversion of cellobiose to glucose were also investigated with the best immobilized biocatalyst.

## 2. Results and Discussion

### 2.1. Use of Different Poly-Functional Molecules and Increased Stability of EaBglA against Temperature

The *Ea*BglA protein was overexpressed in *Escherichia coli* pRARE2 and purified from cell-free enzymatic extract by $Ni^{2+}$ affinity and size-exclusion chromatography. SDS-PAGE loaded with purified

*Ea*BglA displayed an apparent molecular weight of approximately ~52.5 kDa (Figure S1). The yield of purified *Ea*BglA was approximately 18 mg per litre of cell culture.

Two different immobilization approaches were chosen in order to keep the BGL quaternary structure unaltered, in an attempt to maintain the correct assembly of the oligomeric form and thus preserving the enzymatic activity as much as possible. In this sense, glutaraldehyde support (Agarose-GLU) and metal glutaraldehyde-activated support (Agarose-GLU-IDA-$Co^{2+}$) were chosen as a simple non-distorting immobilization method. It is expected that immobilization using Agarose-GLU occurs by non-specific covalent interaction via glutaraldehyde groups, whereas Agarose-GLU-IDA-$Co^{2+}$ will be formed by a specific interaction via his-tagger region, together with a non-specific interaction by glutaraldehyde groups. As shown in Table 1, the recombinant *Ea*BglA was quickly immobilized at pH 7.0 on both supports, with an immobilization efficiency of 100%, occurring in less than 2 h. In order to further improve the stability of its oligomeric form and create an inert coupling enzyme-support, post-immobilization treatments using different poly-functional macromolecules (Tables 1 and 2) were evaluated. Glutaraldehyde-activated supports allow its blocking with compounds containing an amine in its structure. Therefore, by the application of poly-functional macromolecules, it is expected to achieve the blocking of reactive groups available in glutaraldehyde combined with the creation of different microenvironments around the enzyme-support [32–36]. The use of Agarose-GLU support and addition of post-immobilization treatments showed recovered activities from 19 to 56%, while in recovered activities from 8 to 68% were found Agarose-GLU-IDA-$Co^{2+}$ (Table 1). The control preparation (supports without treatments) showed initial recovered activity of 54% in Agarose-GLU and 64% Agarose-GLU-IDA-$Co^{2+}$. The loss of BGL activity can be attributed to amino acids attachments present on its surface (Lys and Arg) and glutaraldehyde groups that inactivate the active site or otherwise sterically hinder enzyme/substrate recognition [37].

To elucidate the efficiency of the post-immobilization step on increasing the structural stability, the thermal stability of all immobilized preparations at 40 °C was compared to free *Ea*BglA. As shown in Table 3, using the immobilized *Ea*BglA on the glutaraldehyde support (Agarose-GLU), it was observed that the *Ea*BglA cross-linking with polyethylenimine and glycine (2.4 and 2.3-fold more stable than free form, respectively) showed the best results. The *Ea*BglA immobilization on Agarose-GLU support without addition of poly-functional macromolecules was performed and demonstrated to be 1.8-fold more stable than free form. In addition, immobilized *Ea*BglA on Agarose-GLU showed a slight improvement in thermal stability than free *Ea*BglA when the treatments with aspartic acid, DEAE-dextran, polygalacturonic acid and a mixture of glycine with polygalacturonic acid were applied (Table 3). At the same conditions, the treatment with the mixture glycine and polyethylenimine showed no significant improve in the thermal stability in relation to the free *Ea*BglA (half-life time of 21 min at 40 °C, Table 3).

In studies with the immobilized *Ea*BglA on the Agarose-GLU-IDA-$Co^{2+}$, the best result was observed by the treatment with polyethylenimine (half-life time of 960 min), which was more stable than the preparation without the addition of poly-functional macromolecules (4.2-fold than control support) and 46-times more stable than free form (half-life time of 21 min at 40 °C, Table 3). All other treatments showed a slight improvement in thermal stability than free *Ea*BglA but no better than control support (immobilized *Ea*BglA without treatments, Table 3).

As noted, the optimal microenvironment formed by post-immobilization of *Ea*BglA with polyethylenimine (PEI) in both supports allowed a tremendous increase in its structural resistance to heat compared to free *Ea*BglA. Literature data report the use of polyethylenimine as advantageous because due to its cationic polymeric nature, it is likely to interact with areas on the protein surface located in different enzyme subunits and is therefore desirable in the case of multimeric enzymes [38]. Thus, the use of PEI-coated supports ensures a strong ionic interaction with the enzyme avoiding distortion of its 3D structure [33,38]. Therefore, due to the excellent result found in the Agarose- GLU-IDA-$Co^{2+}$ immobilization following post-immobilization with PEI to attach and stabilize the tetramer arranged of *Ea*BglA (named, agarose-GLU-IDA-$Co^{2+}$-*Ea*BglA-PEI), it was selected as best system and used in studies of biochemical, sugar and ethanol tolerance, storage and operational stability.

**Table 1.** Immobilization of *Ea*BglA on activated agarose supports followed by post-immobilization with poly-functional macromolecules.

| Support | Treatment | Immobilization Efficiency [a] (%), EI | Recovered Activity after Treatments [b] (%), R |
|---|---|---|---|
| | Control (without treatment) | 100 | 50 |
| Agarose-GLU | Glycine 200 mM, pH 7.0 | 100 | 34 |
| | Aspartic acid 200 mM, pH 7.0 | 100 | 47 |
| | DEAE-Dextran 20 mg mL$^{-1}$, pH 7.0 | 100 | 51 |
| | Polygalacturonic acid 20 mg mL$^{-1}$, pH 7.0 | 100 | 54 |
| | Polyethylenimine 20 mg mL$^{-1}$, pH 7.0 | 100 | 32 |
| | Glycine 200 mM and polygalacturonic acid 20 mg mL$^{-1}$, pH 7.0 | 100 | 56 |
| | Glycine 200 mM and polyethylenimine 20 mg mL$^{-1}$, pH 7.0 | 100 | 19 |
| | Control (without treatment) | 100 | 64 |
| Agarose-GLU-IDA-Co$^{2+}$ | Glycine 200 mM, pH 7.0 | 100 | 63 |
| | Aspartic acid 200 mM, pH 7.0 | 100 | 68 |
| | DEAE-Dextran 20 mg mL$^{-1}$, pH 7.0 | 100 | 8 |
| | Polygalacturonic acid 20 mg mL$^{-1}$, pH 7.0 | 100 | 64 |
| | Polyethylenimine 20 mg mL$^{-1}$, pH 7.0 | 100 | 55 |
| | Glycine 200 mM and polygalacturonic acid 20 mg mL$^{-1}$, pH 7.0 | 100 | 63 |
| | Glycine 200 mM and polyethylenimine 20 mg mL$^{-1}$, pH 7.0 | 100 | 10 |

Co$^{2+}$ (cobalt); IDA (iminodiacetic acid); and GLU (glutaraldehyde). [a] Calculated as the difference between the initial and final activities in the supernatant after 1 h of immobilization. [b] Recovered activity (%), measured as the ratio between the real activity (U g$^{-1}$ of support) of the immobilized *Ea*BglA and theoretical activity of the immobilized *Ea*BglA (U g$^{-1}$ of support).

**Table 2.** Poly-functional macromolecules used in the post-immobilization step and their structural characteristics.

| Poly-Functional Macromolecule | | Structure |
|---|---|---|
| Polygalacturonic acid | Anionic |  |
| Glycine | Zwitterion |  |
| Aspartic acid | Anionic |  |
| DEAE-Dextran | Cationic |  |
| Polyethylenimine | Cationic |  |

**Table 3.** Time course of thermal stability of different immobilized *Ea*BglA derivates.

| Support | Treatment | Half Life ($T_{1/2}$, min 40 °C) | Stability Factor |
|---|---|---|---|
| Free *Ea*BglA | | 21 | |
| Agarose-GLU | Control (without treatment) | 39 | 1.8 |
| | Glycine 200 mM, pH 7.0 | 49 | 2.3 |
| | Aspartic acid 200 mM, pH 7.0 | 26 | 1.2 |
| | DEAE-Dextran 20 mg mL$^{-1}$, pH 7.0 | 25 | 1.2 |
| | Polygalacturonic acid 20 mg mL$^{-1}$, pH 7.0 | 30 | 1.4 |
| | Polyethylenimine 20 mg mL$^{-1}$, pH 7.0 | 50 | 2.4 |
| | Glycine 200 mM and polygalacturonic acid 20 mg mL$^{-1}$, pH 7.0 | 30 | 1.4 |
| | Glycine 200 mM and polyethylenimine 20 mg mL$^{-1}$, pH 7.0 | 22 | 1.0 |
| Agarose-GLU-IDA-Co$^{2+}$ | Control (without treatment) | 230 | 11 |
| | Glycine 200 mM, pH 7.0 | 40 | 1.9 |
| | Aspartic acid 200 mM, pH 7.0 | 23 | 1.1 |
| | DEAE-Dextran 20 mg mL$^{-1}$, pH 7.0 | 28 | 1.3 |
| | Polygalacturonic acid 20 mg mL$^{-1}$, pH 7.0 | 60 | 2.9 |
| | Polyethylenimine 20 mg mL$^{-1}$, pH 7.0 | 960 | 46 |
| | Glycine 200 mM and polygalacturonic acid 20 mg mL$^{-1}$, pH 7.0 | 55 | 2.6 |
| | Glycine 200 mM and polyethylenimine 20 mg mL$^{-1}$, pH 7.0 | 28 | 1.3 |

Co$^{2+}$ (cobalt); IDA (iminodiacetic acid); and GLU (glutaraldehyde). Thermal stability of *Ea*BglA preparations was checked by incubation at 40 °C. Aliquots were periodically withdrawn for quantification of the residual enzymatic activity to estimate the half-life according to Henley and Sadana [39]. The stability factors were done in relation to the free form.

## 2.2. Effects of the Immobilization on the Functional Properties of EaBglA

To better understand how immobilization affects the *Ea*BglA properties, functional characterization of free *Ea*BglA and agarose-GLU-IDA-Co$^{2+}$-*Ea*BglA-PEI was conducted. The pH-dependent enzyme activity, displayed in Figure 1a, showed that the free *Ea*BglA retains its highest relative activity (>80%) between pH 6.0 and 8.0, with a catalytic optimum at 7.0, while agarose-GLU-IDA-Co$^{2+}$-*Ea*BglA-PEI had its catalytic optimum in a broader range of pH values (>80% at pH 3.0–9.0) (Figure 1a). The optimum pH for the agarose-GLU-IDA-Co$^{2+}$-*Ea*BglA-PEI showed a small displacement being found in acid pH values (pH 6.0, 100%). A similar resistance to the pH obtained after the immobilization step was reported in the cellulase immobilization, where it showed that the enzyme was quite stable at pH 1.5 to 12.0 [40]. The catalytic activity of free *Ea*BglA and agarose-GLU-IDA-Co$^{2+}$-*Ea*BglA-PEI was investigated at different temperatures (15 to 70 °C). The optimum reactive temperature of free *Ea*BglA was observed at 30 °C; however, in assays performed with agarose-GLU-IDA-Co$^{2+}$-*Ea*BglA-PEI the highest catalytic activity was presented at 50 °C (Figure 1b). The use of the immobilization step with PEI allowed the *Ea*BglA to expand its catalytic performance at higher temperatures, relative activity R > 50% between 30–50 °C. The improved resistance against temperature can be attributed to the conformational rigidity of β-glucosidase after immobilization. A similar phenomenon had been reported. For example, Samak et al. prepared magnetic graphene oxide (MGO) nano-sheets that were used to immobilize His-tagged CotA laccase [41]. In their experiments, the immobilized laccase showed optimal activities at higher temperatures compared to the free form [41]. Driss et al. applied Ni$^{2+}$-chelate Eupergit C to immobilize his-tagged xylanase from *Penicillium occitanis* reached elevations of 15 °C [42].

Considering the enzymatic properties sought for biotechnological applications, stability for prolonged periods of time in a broad pH range is usually amongst the most frequently targeted biochemical characteristics. Therefore, stability of *Ea*BglA as a function of pH was studied. The results for pH-stability using free *Ea*BglA and agarose-GLU-IDA-Co$^{2+}$-*Ea*BglA-PEI were evaluated at different pH values ranging from 3.0 to 10 (Figure 1c,d). After 72 h, the agarose-GLU-IDA-Co$^{2+}$-*Ea*BglA-PEI showed to be stable over a relatively broad pH range, that is between pH 5.0–9.0, as it retained 50–100% initial activity (Figure 1c). Differently, free *Ea*BglA had a low activity after 24 h at all pH values tested (Figure 1d). Thus, compared to the original free enzyme, the immobilized β-glucosidase could work in harsh environmental conditions with less activity loss. Stabilities in a pH range can be an important factor for simultaneous saccharification and fermentation (SSF) process, in which the ethanol fermentation performed by *Saccharomyces cerevisiae* occurs at pH 4.0–5.5 [43].

From the economic point of view, the microenvironment formed by the post-immobilization step with PEI has now enabled the immobilized BGL to expand its uses in mesophilic reactions, that is, commercial cocktails used for the saccharification of lignocellulosic biomass [3,44,45] and also in processes that require green catalysts that are more resistant to pH [43].

## 2.3. Effect of Immobilization on the Resistance to Glucose and Ethanol

To investigate the suitability of agarose-GLU-IDA-Co$^{2+}$-*Ea*BglA-PEI catalysed processes in industrial green reactions, the effects of inhibitors were analysed (Figure 1e,f). The prevalence of glucose and ethanol at higher concentrations may cause an inhibitory effect on β-glucosidases [6]. Since these coproducts are naturally obtained during saccharification and fermentation, BGLs that tolerate high concentrations of these components would be essential for an efficient fermentation process. BGL from the psychrophilic bacterium *E. antarcticum* B7 was reported previously as having a high tolerance to glucose inhibition (retaining 50% relative activity at 1 M glucose) [11]. The properties showed by *Ea*BglA are extremely desirable for biocatalysts involved with the breakdown of polysaccharides such as cellulose or cellobiose into glucose [4]. Glucose inhibition in agarose-GLU-IDA-Co$^{2+}$-*Ea*BglA-PEI was not affected by glucose up to 100 mM when *p*NPG was used as a substrate (113% relative activity, Figure 1e). The agarose-GLU-IDA-Co$^{2+}$-*Ea*BglA-PEI retained 55% relative activity in the presence of 600 mM glucose and when 1 M glucose was applied, immobilized

β-glucosidase retained around 27% of its initial activity (Figure 1e). As shown, glucose inhibition of the immobilized *Ea*BglA was not dramatically affected and therefore, it could be adopted as a viable biocatalyst in sustainable processes of cellulosic biomass degradation, such as SSF process.

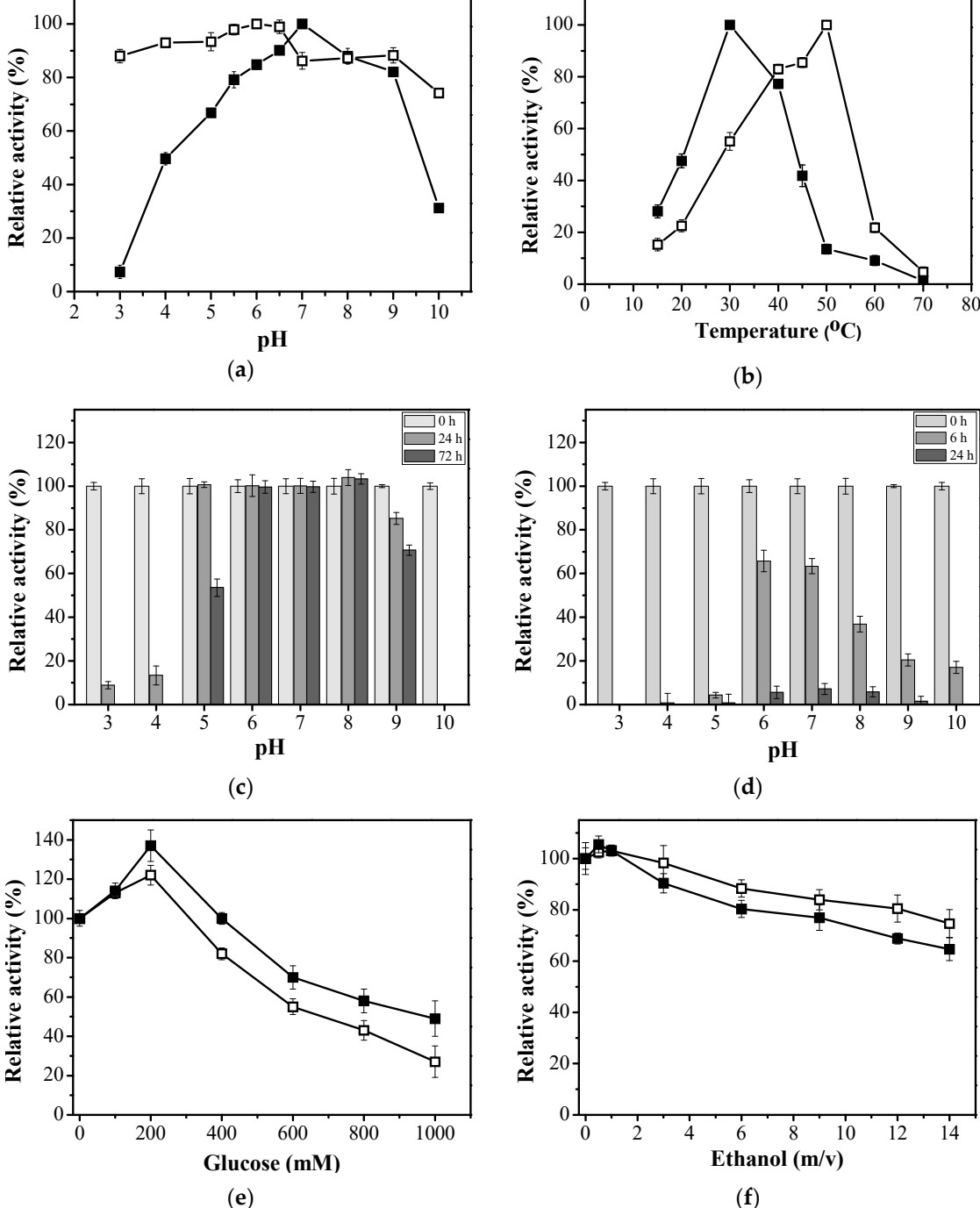

**Figure 1.** Functional analysis of free *Ea*BglA (filled squares) and agarose-GLU-IDA-Co$^{2+}$-*Ea*BglA-PEI (empty squares); (**a**) Effect of pH and, (**b**) temperature on the enzymatic activity. pH-stability under different pH values; (**c**) agarose-GLU-IDA-Co$^{2+}$-*Ea*BglA-PEI and (**d**) free *Ea*BglA. (**e**) The effect of glucose (0–1500 mM) and, (**f**) ethanol (0–14%, *v/v*) on BGL activity. All enzymatic properties were determined using *p*NPG as substrate. All measurements were done in triplicates. Error bars show SD. The initial activities were regarded as 100% and relative activities were expressed as a percentage of initial activity.

Notably, the effect of ethanol on enzymatic activity is essential for β-glycosidase characterizations, since these enzymes are exposed to substantial concentrations of ethanol in a number of applications [46–48]. To investigate its effects on free *Ea*BglA and agarose-GLU-IDA-Co$^{2+}$-*Ea*BglA-PEI, ethanol at various concentrations was used (Figure 1f). Ethanol concentrations up to 1% (*v/v*) potentiated the enzymatic activity in both cases. Increasing ethanol concentration to 14% significantly reduced enzymatic activity of free *Ea*BglA to ~68%, while the agarose-GLU-IDA-Co$^{2+}$-*Ea*BglA-PEI had a slight loss of activity (retained ~80% of initial activity) under the conditions of analysis (Figure 1f). Considering that in traditional processes the final ethanol concentration in fermented broths is around 10% [46], it is inferred that the immobilized *Ea*BglA is fully compatible with typical *S. cerevisiae* fermentation processes.

### 2.4. Conversion of Cellobiose to Glucose and Storage

Unlike free enzymes, recycling of their immobilized counterparts offers technical and economic advantages in industrial bioreactors [14,49]. To assess the operational stability, immobilized *Ea*BglA was applied in successive reactions using cellobiose as substrate. After each reaction cycle (30 min), the immobilized BGL was thoroughly washed several times with sodium phosphate buffer (50 mM at pH 7.0) to remove residual substrate and/or product formed. As illustrated in Figure 2a,b, after 9 cycles, agarose-GLU-IDA-Co$^{2+}$-*Ea*BglA-PEI retained approximately 100% of its initial activity with a yield of 100%. Therefore, the post-immobilization applied in the work provided a more suitable biocatalyst, thus reducing processing costs in practical applications. Besides, some studies describe that the activity of the enzyme treated with glutaraldehyde can withstand after several cycles of reuse by various investigators [50–55].

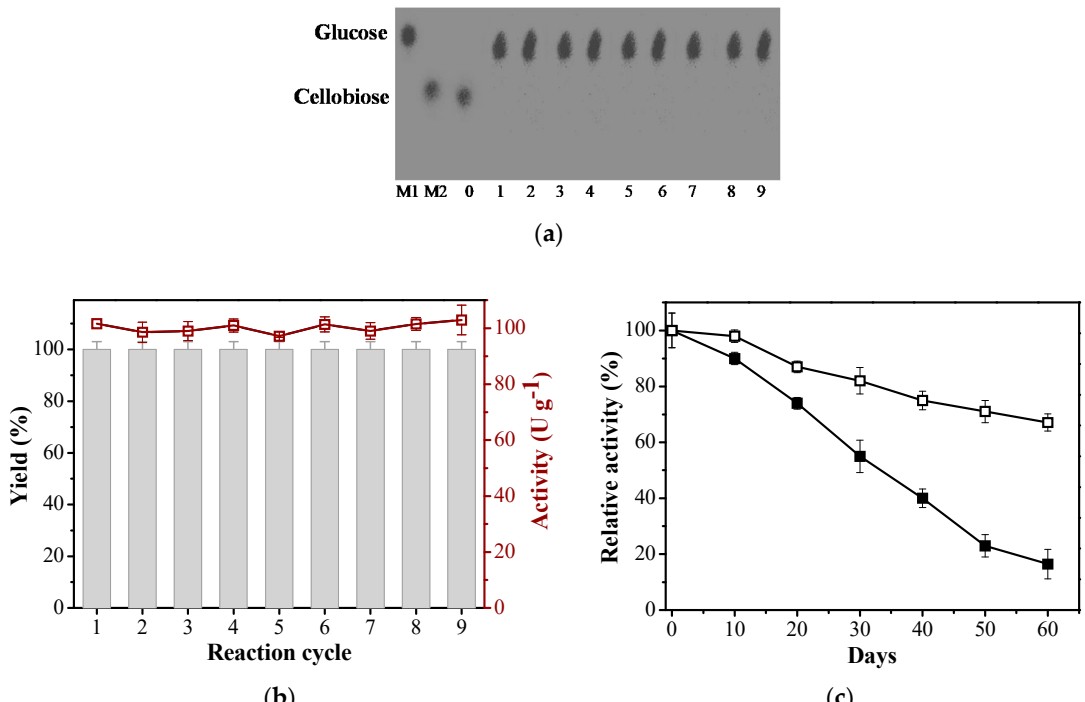

**Figure 2.** (**a**) Thin-layer chromatography of sugars produced from hydrolysis enzymatic in different recycle. M1: glucose, M2: cellobiose, lane 0, 1, 2, 3, 4, 5, 6, 7, 8, 9: cellobiose incubated with agarose-GLU-IDA-Co$^{2+}$-*Ea*BglA-PEI (167 U g$^{-1}$ of support). (**b**) Reusability assay of agarose-GLU-IDA-Co$^{2+}$-*Ea*BglA-PEI using cellobiose as substrate. Line (activity U g$^{-1}$ of support) and bars (yield %). (**c**) Storage stability of free and agarose-GLU-IDA-Co$^{2+}$-*Ea*BglA-PEI (storage conditions: 4 °C, pH 7.0). Free *Ea*BglA (filled squares) and agarose-GLU-IDA-Co$^{2+}$-*Ea*BglA-PEI (empty squares). The initial activities were regarded as 100% and relative activities were expressed as a percentage of initial activity.

Good storage stability is another essential property for an economically-feasible biocatalyst [56]. Both free *Ea*BglA and agarose-GLU-IDA-Co$^{2+}$-*Ea*BglA-PEI were stored in sodium phosphate buffer (50 mM, pH 7.0). Agarose-GLU-IDA-Co$^{2+}$-*Ea*BglA-PEI kept 67.1% of its original activity, while the free BGL retained 16.4% of its activity at 4 °C after 60 days (Figure 2c). It suggests that the mechanical rearrangement provided by the immobilization with flexible cationic polymer (PEI) can maintain the *Ea*BglA in a stable state compared to its free form.

## 3. Materials and Methods

### 3.1. Material

Agarose 4 BCL was purchased from Agarose Bead Technologies (Madrid, Spain). Epichlorohydrin, iminodiacetic acid, sodium borohydride, sodium periodate, bovine serum albumin, polyethylenimine (PEI-25kDa), isopropyl-β-D-thiogalactopyranoside (IPTG), 4-nitrophenyl β-D-glucopyranoside (*p*NPG) were purchased from Sigma (Sigma-Aldrich®, St Louis, MO, USA). Glutaraldehyde and ethylenediamine were purchased from Alfa Aesar (Thermo Fisher Scientific®, Waltham, MA, USA). The protein assay kit (Bio-Rad protein dye reagent concentrate) was sourced from Bio-Rad. Superdex-200 column and His-Trap Ni$^{2+}$-chelating affinity were purchased from GE Healthcare (Pittsburgh, PA, USA). The silica gel TLC plates (DC Kieselgel 60 F254) was purchased from Merck® (Kenilworth, NJ, USA). Expression vector pET28a(+) was purchased from Novagen® (Thermo Fisher Scientific®, Waltham, MA, USA). *Escherichia coli* strains DH5α and pRARE2 (Agilent Technologies, Santa Clara, CA, USA) were used as cloning and expression hosts, respectively. All other reagents and solvents are of analytical grade and commercially available.

### 3.2. Cloning, Heterologous Expression and Purification Protein

The putative β-glucosidase gene from *Exiguobacterium antarcticum* B7 (BglA) was cloned into pET28a(+) vector with a hexahistidine-tag at the N-terminus as described by a previous protocol Crespim et al [11]. Then, *Escherichia coli* pRARE2 cells carrying the pET28a(+)/BglA were plated in selective solid Luria-Bertani (LB) agar containing kanamycin (30 mg mL$^{-1}$) and chloramphenicol (30 mg mL$^{-1}$). Cells from a single colony were cultivated into LB medium in a shaker (New Brunswick Scientific, New Jersey, USA) at 250 rpm, 37 °C. When the OD$_{600}$ value of the culture reached 0.6–0.8, a final concentration of 0.5 mM isopropyl-β-D-thiogalactopyranoside (IPTG) was added to induce the expression. The culture with IPTG was incubated for 4 h before harvesting the cells by centrifugation at 7000× *g* (20 min, 4 °C). The cell pellet was resuspended in lysis buffer (25 mM sodium phosphate buffer, 300 mM NaCl, 20 mM imidazole, pH 7.5) supplemented with 1 mM PMSF and lysed by 0.1 mg mL$^{-1}$ lysozyme treatment followed by sonication. The cell extracts were clarified by centrifugation (12,000× *g*, 30 min, 4 °C) and the supernatant was applied onto 5 mL Hi-Trap chelating HP column (GE Healthcare Biosciences, Pittsburgh, PA, USA) coupled to an ÄKTA system (GE Healthcare Biosciences, Pittsburgh, USA) and pre-equilibrated with 25 mM sodium phosphate, 300 mM NaCl and 20 mM imidazole, pH 7.5. The His-tagged protein (*Ea*BglA) was then eluted with a nonlinear gradient of buffer B (25 mM sodium phosphate, 300 mM NaCl, 500 mM imidazole, pH 7.5). The fractions containing β-glucosidase activities were pooled, concentrated using Amicon Ultra-4 30 K centrifugal filter units (Millipore, Burlington, MA, USA) and submitted to size-exclusion chromatography using a HiLoad 16/60 Superdex 200 pg column (GE Healthcare Biosciences, Pittsburgh, PA, USA) previously equilibrated with 25 mM sodium phosphate, 150 mM NaCl, pH 7.5, at a flow rate of 1.0 mL min$^{-1}$. The purity of the final β-glucosidase was verified by sodium dodecyl sulphate-polyacrylamide gel electrophoresis (SDS-PAGE, 12%). The protein concentration was determined by absorbance at 280 nm using the molar extinction coefficient (110,030 M$^{-1}$ cm$^{-1}$) obtained from the amino acid composition and, Bradford method's [57] using a Protein Assay Kit (Bio-Rad protein dye reagent concentrate) with bovine serum albumin as the standard.

### 3.3. Construction of Supports Applied in the EaBglA Immobilization

#### 3.3.1. Activation of Agarose-Based Beads

Epoxy-activated agarose was prepared by the reaction of hydroxyl groups on the agarose beads with epichlorohydrin, as described by Mateo et al [58]. Agarose gel 4 BCL (50 g) was washed with excess distilled water and then treated with 220 mL of NaOH 1.84 M under agitation and in an ice bath. Subsequently, 1.0 g of sodium borohydride ($NaBH_4$), 80 mL of acetone and 55 mL of epichlorohydrin were added and the suspension formed was gently stirred at 25 °C for 16 h. Epoxy agarose support was then washed five times with distilled water and filtered under vacuum. After the last washing, the support was thoroughly sucked dry to remove the interstitial humidity and stored at 4 °C.

The number of epoxy/ligand groups was characterized as described Whistler et al. [59] and indicated a total amount of activated primary hydroxyl groups around $65 \pm 0.3$ µmoL g$^{-1}$, with epoxy groups accounting for $23 \pm 0.4$ µmoL g$^{-1}$ and, diol groups accounting for $42 \pm 0.4$ µmoL g$^{-1}$.

#### 3.3.2. Modification of Epoxy-Activated Agarose Beads

##### Support Activated with Glutaraldehyde Group

Activation by glutaraldehyde groups (Agarose-GLU) was prepared as previously described by Betancor et al [60]. Briefly, the epoxy agarose beads (10 g) were hydrolysed with 100 mL of 1 M $H_2SO_4$ for 2 h at 25 °C. Afterwards, the support was washed with distilled water and oxidized using 100 mL of $NaIO_4$ (100 mM). Then, the support was treated with 100 mL of 2 M ethylenediamine (EDA) at pH 10 and kept under gentle stirring at 25 °C. After 2 h, sodium borohydride (10 mg mL$^{-1}$) was added and stirred for further 2 h at 25 °C. The particles were successively washed with distilled water and 11 mL of glutaraldehyde solution (25%, *v/v*) was added together with 17 mL of sodium phosphate buffer (200 mM at pH 7.0). The system was kept under gentle stirring for 18 h at 25 °C. Finally, the activated support was washed with distilled water and vacuum dried.

##### Anionic Support Activated with Glutaraldehyde Group and Metal Chelate

Agarose support (10 g) activated with the glutaraldehyde group and metal chelate (Agarose-GLU-IDA-Co$^{2+}$) was obtained by treatment of epoxy agarose beads with 100 mL of 0.5 M iminodiacetic acid (IDA) at pH 11 for 18 h at 25 °C. Then, the support was washed with distilled water and oxidized using 100 mL of sodium periodate ($NaIO_4$) at a final concentration of 100 mM. After 2 h of gentle agitation at 25 °C, the oxidized support was washed with distilled water and treated with 100 mL of 2 M ethylenediamine (EDA) pH 10 for 2 h at 25 °C. Sodium borohydride (10 mg mL$^{-1}$) was added and stirred for further 2 h at 25 °C. The activated agarose was successively washed with distilled water. Glutaraldehyde solution (25%, *v/v*) and sodium phosphate buffer (200 mM at pH 7.0) were added and the system was kept under gentle stirring for 18 h at 25 °C. The agarose activated with glutaraldehyde was added to the metal chelate solutions (30 mg mL$^{-1}$, $CoCl_2$) at pH 7 for 30 minutes at 25 °C [31]. The activated support was washed with distilled water and vacuum dried.

### 3.4. Immobilization of EaBglA on Activated Agarose Supports

The immobilization course of β-glucosidase (*Ea*BglA) was monitored measuring the enzyme activity in the supernatant and in the whole suspension at different time intervals. Additionally, controls with free *Ea*BglA were used to determine a possible inactivating effect (pH, temperature or dilution) on the enzyme during the immobilization process. In all cases, β-glucosidase was diluted in sodium phosphate buffer (200 mM at pH 7.0) and the suspension was gently stirred during 10 min. Then, different immobilization supports were suspended in an enzyme solution: 1 g agarose beads to 10 mL enzyme solution in immobilization buffer (~7.0 U g$^{-1}$ of support) and gently stirred at 25 °C at different times. The immobilization was considered complete when there was not activity in the supernatant.

After immobilization step, poly-functional macromolecules (i.e., DEAE-Dextran, polygalacturonic acid, polyethylenimine, aspartic acid and glycine) were evaluated in a post-immobilization to stabilize the *Ea*BglA. The reactions were maintained under agitation for 1 h (glycine reaction) and 30 min (all the others) using sodium phosphate buffer (200 mM at pH 7.0).

The immobilization efficiency (IE, %) was calculated as Equation (1):

$$\text{IE} = \frac{A_i - A_f}{A_i} \times 100\% \tag{1}$$

where $A_i$ is the initial enzymatic activity (U) of the *Ea*BglA prior to immobilization and $A_f$ is the enzymatic activity (U) remaining in the supernatant at the end of the immobilization procedure.

The recovered activity (R, %) was calculated as Equation (2):

$$R = \frac{A_o}{A_T} \times 100\% \tag{2}$$

where $A_o$ is the as the ratio between the real activity (U $g^{-1}$ of support) of the immobilized preparation and $A_T$ is the theoretical activity (U $g^{-1}$ of support) of the immobilized preparation.

### 3.5. Functional Assays

Enzymatic assays were performed to characterize the functional proprieties of free and immobilized *Ea*BglA. Thus, all activities were measured by increase in absorbance at 410 nm produced by the release of *p*-nitrophenol (*p*NP) in the hydrolysis of 4-nitrophenyl β-D-glucopyranoside (*p*NPG, 0.5 mM) in sodium phosphate buffer (50 mM, pH 7.0) ($\varepsilon_{410\,nm,\,pH\,7}$ = 7320 $M^{-1}$ $cm^{-1}$). One unit of activity (U) was defined as the production of 1 µmoL of *p*-nitrophenol per minute, under the assay conditions and, the specific activity was defined as the number of units per mg of protein [11].

The effect of pH on the activity was determined using the synthetic substrate 4-nitrophenyl β-D-glucopyranoside (*p*NPG) in the range 3.0–10 at 30 °C, using 50 mM of McIlvaine glycine-supplemented buffer. The optimum temperature of enzyme activity was determined in assays ranging from 15 to 70 °C. For pH tolerance tests, free and immobilized *Ea*BglA were evaluated in optimal conditions using pH range from 3.0 to 10.0 using McIlvaine glycine-supplemented buffer at 50 mM. Thermal stability was evaluated by pre-incubating the free and immobilized *Ea*BglA in sodium phosphate buffer (50 mM at pH 7.0) at 40 °C. Thermal inactivation was calculated based on the deactivation theory proposed by Henley and Sadana [39]. Inactivation parameters were determined from the best-fit model of the experimental data which was the one based on a two-stage series inactivation mechanism with residual activity. Half-life was used to compare the stability of the different preparations, being determined by interpolation from the respective models described in Addorisio et al. [61]. Periodically, samples of these suspensions were withdrawn and the remaining activities assayed. The initial activities were regarded as 100% and relative activities were expressed as a percentage of initial activity. All assays were performed in triplicate.

Stability in high glucose concentrations, up to 1000 mM and ethanol (0–25%, *v/v*) was investigated in optimal conditions by incubating *Ea*BglA in 50 mM sodium phosphate buffer (pH 7.0). The residual activity was evaluated using *p*NPG as the substrate as described above.

### 3.6. Operational Stability and Storage

The recycling capacity of *Ea*BglA was determined by measuring its activity in 9 consecutive cycles. For these experiments, 100 mg immobilized *Ea*BglA (167 U $g^{-1}$ of support) was added to a suspension containing cellobiose (10%, *w/v*) in sodium phosphate buffer (50 mM at pH 7.0) and after a 30 min reaction at 50 °C, the immobilized catalyst was collected by centrifugation. After each cycle, the immobilized *Ea*BglA was washed three times with assay buffer, performed its activity and added to a novel cellobiose solution. Analysis of cellobiose hydrolysis was performed by ascending

chromatography on a thin layer of silica (DC Kieselgel 60 F254). Aliquots were sampled and applied to a silica plate. The run was carried out with a mixture of n-butanol: acetic acid: distilled water (2:1:1 *v/v*). The revelation of sugars was performed by dipping the plate in 5% sulfuric acid in ethanol and heating.

Storage stability was tested using free and immobilized BGL in sodium phosphate buffer (50 mM at pH 7.0) for 60 days at 4 °C. The activity was measured with the *p*NPG assay.

## 4. Conclusions

An ideal β-glucosidase for industrial applications needs to be a recyclable biocatalyst with high catalytic efficiency, excellent thermostability and resistance to product inhibition. In this work, β-glucosidase (*Ea*BglA) immobilized on metal glutaraldehyde-activated agarose support following by cross-linking with polyethylenimine proved to be an excellent system to attach and stabilize the BGL. Surprisingly, the optimal microenvironment formed by the crosslinking with PEI made *Ea*BglA more resistant to pH and temperature, increasing its thermal and storage stability. Additionally, it provided retention of more than 100% of its initial activity after 9 cycles of cellobiose hydrolysis. Therefore, the technique applied to BGL immobilization has contributed to the production of a new and more stable enzyme, which can expand its application, having a good perspective in areas such as biofuels.

**Supplementary Materials:** The following are available online at http://www.mdpi.com/2073-4344/9/3/223/s1, Figure S1: SDS-PAGE of the purified EaBglA - M, molecular marker (kDa); 1: *E. coli* pRARE2 crude extract with *Ea*Bgl; 2: *Ea*BglA after purification by His-Trap Ni$^{2+}$-chelating affinity column and size-exclusion chromatography.

**Author Contributions:** R.R.d.M., R.C.A. and A.S.d.S. performed the experiments and partially wrote the paper; R.R. and H.H.S. prepared the enzyme and partially wrote the paper; C.M. conceived and designed the experiments and partially wrote the paper.

**Acknowledgments:** The authors gratefully acknowledge the Ramón Areces Foundation and Spanish Government (AGL2017-84614-C2-1-R) for financial support and National Counsel of Technological and Scientific Development-CNPq (429829/2016-7). R.R.M. was supported by FAPESP fellowship (2017/14253-9). We thank the Brazilian Bioethanol Science and Technology Laboratory CTBE/LAM Facility.

**Conflicts of Interest:** The authors declare no conflict of interest.

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
