# Peer review of "Cross-Linking with Polyethylenimine Confers Better Functional Characteristics to an Immobilized β-glucosidase from Exiguobacterium antarcticum B7"

_catalysts, doi:10.3390/catal9030223_

Round 1
Reviewer 1 Report
The manuscript described the evaluation of a recombinant beta-glucosidase from Exiguobacterium antircticum B7 (recEaBglA) for biocatalysis. EaBglA is a tetrameric endoglucanase, and the authors explored the immobilization of the enzyme on activated agarose with the aim at increasing stability and enzymatic activity for use in biocatalysis. The authors explore two main immobilization media, that being glutaraldehyde or metal-chelated glutaraldehyde modified agarose, agarose-GLU and agarose-GLU-IDA-Co, respectively. They also explore a series of surface modifications post-recEaBglA immobilization in an effort to stabilize the structures. The authors identify that recEaBglA immobilized on agarose-GLU-IDA-Co2+ with subsequent polyethylimine (PEI) surface modification provides the best relative enzymatic activity with significant increase in stability. They further characterize is composite (recEaBglA-agarose-GLU-IDA-Co2+-PEI) for retention of catalytic activity, range of activity and overall stability.
The manuscript is overall well-constructed and the results and conclusions reasonably presented. There are some issues to be addressed, which I outline below:
- On Line 95, the authors state “histidine richest place” for non-specific binding of the protein to the modified agarose surface. This really should be “the histidine tag”, because this is the most likely interaction. The authors use a His-tagged protein and purify using immobilized metal chromatography (HiTrap chelating) with Cu2+ as the bound ion on the resin. The authors do not remove the His-tag of the protein following purification (they don’t state it in the methods), so this is the likely point of interaction with any immobilized metal ions on the agarose support, not an unspecified histidine patch, which I don’t recall on any of the Bgl structures in the Protein Data Bank that I’ve surveyed.
- Following on the above point, the authors indicate that their loss of observed activity upon recEaBglA immobilization onto the activated agarose is likely due to attachment by “different amino acids” (Lines 108-110). Surely the authors mean different amines, such as Lys vs. Arg sidechains, as glutaraldehyde is reactive to amines? What you have in this case is a non-specific interaction on the Agarose-GLU (via free side chain amines) versus a specific interaction on the IDA-Co2+, this being via the His-Tag of the protein. The authors use immobilized metal affinity capture of the recEaBglA protein, which relies on this exact principle. The Co2+ ions provide a stronger binding to the Histidine tag versus that of Cu2+ ions, generally used in preloaded IMAC columns from GE Healthcare.
- General loss in activity is probably a proximity of access of the active sites to the support, and the specific interaction gives greater spatial separation.
- Lines 213 and 241 (Figure legends): Add “recEaBglA” after “free” to clarify for the reader.
- I found the naming of the final composite name a bit misleading. It really should be “agarose-GLU-IDA-Co2+-recEaBglA-PEI” to properly reflect the order in which the components are added to the modified agarose support.
- There are multiple instances where the English grammar and sentence construction should be carefully reviewed and updated. This starts with the abstract, Line 21, and the sentence beginning “Best result…”; clearly this should read “The best result…”. There are many such occurrences of this throughout the manuscript. The authors need to carefully review the manuscript for all these instances and address them.
- Line 49: “discovered should be “found”
- Line 60: “feature” should be “features”
Author Response
Ref manuscript #447125 - Cross-linking with polyethylenimine confers better functional characteristics to an immobilized β-glucosidase from Exiguobacterium antarcticum B7
Dear editor and reviewers,
We would like to sincerely thank you for your time and constructive comments, which very valuable to improve the quality of our manuscript. We acknowledge that some parts of the text to be rewritten aiming at a better presentation of our finding and it was carefully revised as pointed out by the reviewers.
We believe that with the extensive text revision all concerns raised by the reviewers were properly addressed. Please find below our detailed answer to the reviewers’ comments.
Kind regards,
Cesar Mateo, and behalf of all authors.
Reviewer #1
The manuscript described the evaluation of a recombinant beta-glucosidase from Exiguobacterium antircticum B7 (recEaBglA) for biocatalysis. EaBglA is a tetrameric endoglucanase, and the authors explored the immobilization of the enzyme on activated agarose with the aim at increasing stability and enzymatic activity for use in biocatalysis. The authors explore two main immobilization media, that being glutaraldehyde or metal-chelated glutaraldehyde modified agarose, agarose-GLU and agarose-GLU-IDA-Co2+, respectively. They also explore a series of surface modifications post-recEaBglA immobilization in an effort to stabilize the structures. The authors identify that recEaBglA immobilized on agarose-GLU-IDA-Co2+ with subsequent polyethylenimine (PEI) surface modification provides the best relative enzymatic activity with significant increase in stability. They further characterize is composite (recEaBglA-agarose-GLU-IDA-Co2+-PEI) for retention of catalytic activity, range of activity and overall stability. The manuscript is overall well-constructed and the results and conclusions reasonably presented. There are some issues to be addressed, which I outline below:
Comment 1: On Line 95, the authors state “histidine richest place” for non-specific binding of the protein to the modified agarose surface. This really should be “the histidine tag”, because this is the most likely interaction. The authors use a His-tagged protein and purify using immobilized metal chromatography (HiTrap chelating) with Cu2+ as the bound ion on the resin. The authors do not remove the His-tag of the protein following purification (they don’t state it in the methods), so this is the likely point of interaction with any immobilized metal ions on the agarose support, not an unspecified histidine patch, which I don’t recall on any of the Bgl structures in the Protein Data Bank that I’ve surveyed.
Reply 1: We agreed to the reviewer's suggestion. The region with greater probability of connection to the support containing metal group (Agarose-GLU-IDA-Co2+) and the hist-tag region. We include this information in the text so there are no future questions.
Comment 2: Following on the above point, the authors indicate that their loss of observed activity upon recEaBglA immobilization onto the activated agarose is likely due to attachment by “different amino acids” (Lines 108-110). Surely the authors mean different amines, such as Lys vs. Arg side chains, as glutaraldehyde is reactive to amines? What you have in this case is a non-specific interaction on the Agarose-GLU (via free side chain amines) versus a specific interaction on the IDA-Co2+, this being via the His-Tag of the protein. The authors use immobilized metal affinity capture of the recEaBglA protein, which relies on this exact principle. The Co2+ ions provide a stronger binding to the Histidine tag versus that of Cu2+ ions, generally used in preloaded IMAC columns from GE Healthcare.
Reply 2: Yes, the covalent immobilization on glutaraldehyde was performed by amine groups. In the case of the immobilization on Agarose-GLU considering that we are using 200 mM phosphate, the immobilization mainly is oriented through the terminal amine group and then, once immobilized probably more linkages via other amine groups are promoted. In the other case, the immobilization was performed using Co2+ activated supports and purification on Ni2+ activated columns. Applying a specific interaction on the IDA-Co2+ is expected to be performed through the His-tag of the protein. We better organize the sentence in the text so that a false interpretation does not occur.
Comment 3: General loss in activity is probably a proximity of access of the active sites to the support, and the specific interaction gives greater spatial separation.
Reply 3: Probably, the loss of activity is caused by the covalent multi-interaction protein-support. This interaction may be able to distort the enzyme 3-D structure or considering that it is a multimeric enzyme is more probably by disassembling of the subunits of the enzyme. In the global study, other supports were used for the immobilization of this enzyme such as epoxy or aldehyde activated supports (Melo, R.R.; Alnoch, R.C.; Vilela, A.F.L.; Souza, E.M.; Krieger, N.; Ruller, R.; Sato, H.H.; Mateo, C. New heterofunctional supports based on glutaraldehyde-activation : A tool for enzyme immobilization at neutral pH. Molecules. 2017; 22: 1088–106). All these supports have as common characteristics the establishment of more intense multi-interaction enzyme-support processes. This indicates that the loss of activity is not produced by the orientation and more probably by the multi-interaction capable to promote the disassembling of the subunits. This is also coherent with the fact of after using polyfunctional polymers (PEI) the stability was higher than when other reagents were used.
Comment 4: Lines 213 and 241 (Figure legends): Add “recEaBglA” after “free” to clarify for the reader.
Reply 4: We include this information in the manuscript figure 1 and 2 legend.
Comment 5: I found the naming of the final composite name a bit misleading. It really should be “agarose-GLU-IDA-Co2+-recEaBglA-PEI” to properly reflect the order in which the components are added to the modified agarose support.
Reply 5: We thank the Reviewer for this suggestion and agree that naming the final construction a bit misleading. The name ‘‘recEaBglA-GLU-IDA-Co2+-PEI’’ was modified to ‘‘agarose-GLU-IDA-Co2+-EaBglA-PEI’’.
Comment 6: There are multiple instances where the English grammar and sentence construction should be carefully reviewed and updated. This starts with the abstract, Line 21, and the sentence beginning “Best result…”; clearly this should read “The best result…”. There are many such occurrences of this throughout the manuscript. The authors need to carefully review the manuscript for all these instances and address them.
Reply 6: We are thankful for the constructive comment. This new version was carefully revised the English in order to improve the overall quality.
Comment 7: Line 49: “discovered” should be “found”
Reply 7: We corrected in the text.
Comment 8: Line 60: “feature” should be “features”
Reply 8: We corrected in the text.
Reviewer 2 Report
The authors of the presented manuscript studied the immobilization of a glucosidase using two different methods: covalent linkage by glutaraldehyde and affinity immobilization using the his-tag of the enzyme. Furthermore, the immobilized biocatalysts is treated with different polymers or other blocking reagents to study potential improvements of activity/stability. In general, I am very satisfied with the paper and I don't have too many comments. However, I would like to encourage the authors to consider the following comments:
The introduction could conatin a bit more examples of other studies using covalent immobilization of glucosidases as well as utilizing the his-tag for immobilization.
The discussion then could contain a bit more comparison to other studies using similar immobilization techniques on the similar enzymes.
p2, l90-p3, l96: First of all, when using glutaraldehyde for immobilization it is covalent immobilization, so please call it also like that. Then glutaraldehyde is not leading to orientation based on hydrophobicity, it is binding to amino groups present on the enzyme surface. Furhtermore: Why not stating his-tag instead "histidine richest place"? It is most probable the his-tag that leads to immobilization ion the GLU-IDA-Co2+ resin. The other formulations are only confusing the reader.
Table 3: how were the stability factors calculated?
p7, l187-p8, l212: I don't see a significantly higher tolerance towards glucose and ethanol when comparing the free and the immobilized ones. They are rather in a comparable range. Towards glucose the tolerance of the immobilized enzyme even seems to be a bit lower. If you insist on using the word significant, please perform proper statistical test on the data presented in figure 1e+f.
Please give for all figures using relative activities information on what 100% activity is in U/g in the legend. Makes it easier to compare among the different figures.
Author Response
Reviewer #2
The authors of the presented manuscript studied the immobilization of a glucosidase using two different methods: covalent linkage by glutaraldehyde and affinity immobilization using the his-tag of the enzyme. Furthermore, the immobilized biocatalysts is treated with different polymers or other blocking reagents to study potential improvements of activity/stability. In general, I am very satisfied with the paper and I don't have too many comments. However, I would like to encourage the authors to consider the following comments:
Comment 1: The introduction could contain a bit more examples of other studies using covalent immobilization of glucosidases as well as utilizing the his-tag for immobilization.
Reply 1: Literature data reporting different strategies for the immobilization of BGLs were included in introduction part, as suggested by the Reviewer.
Comment 2: The discussion then could contain a bit more comparison to other studies using similar immobilization techniques on the similar enzymes.
Reply 2: We have included, in this new version, more details to enrich the discussion of the proposed material, as suggested by the Reviewer.
Comment 3: p2, l90-p3, l96: First of all, when using glutaraldehyde for immobilization it is covalent immobilization, so please call it also like that. Then glutaraldehyde is not leading to orientation based on hydrophobicity, it is binding to amino groups present on the enzyme surface. Furthermore: Why not stating his-tag instead "histidine richest place"? It is most probable the his-tag that leads to immobilization ion the GLU-IDA-Co2+ resin. The other formulations are only confusing the reader.
Reply 3: Apart from the structural complexity presented by glutaraldehyde, its configuration already characterizes it as having distinct heterofunctionalities. Depending on the conditions applied during the process of immobilization with glutaraldehyde-activated supports, the proteins can be immobilized by three different mechanisms. On using very high ionic strength, the protein may first be immobilized by way of hydrophobic adsorption before forming the covalent bonds, whereas when using low ionic strength, primary immobilization will be by way of anion exchange. If the ionic strength applied were moderate, protein immobilization is described as being mainly due to covalent bonds (Barbosa, O.; Ortiz, C.; Berenguer-Murcia, Á.; Torres, R.; Rodrigues, R.C.; Fernandez-Lafuente, R. Glutaraldehyde in bio-catalysts design: A useful crosslinker and a versatile tool in enzyme immobilization. RSC Adv. 2014, 4, 1583–1600). We agree with the reviewer that BGL immobilization using glutaraldehyde groups with moderate ionic strength (applied at work) occur by direct covalent attachment. So we rewrite the sentence more clearly.
Comment 4: Table 3: how were the stability factors calculated?
Reply 4: Thermal inactivation was calculated based on the deactivation theory proposed by Henley and Sadana (Henley, J.P.; Sadana, A. Deactivation theory. Biotechnol. Bioeng. 1986, 28, 1277–1285). Inactivation parameters were determined from the best-fit model of the experimental data which was the one based on a two-stage series inactivation mechanism with residual activity. Half-life was used to compare the stability of the different preparations, being determined by interpolation from the respective models described in (Addorisio, V.; Sannino, F.; Mateo, C.; Guisan, J.M. Oxidation of phenyl compounds using strongly stable immobilized-stabilized laccase from Trametes versicolor. Process Biochem. 2013, 48, 1174–1180). We include this information in the manuscript in item 3.5. Functional assays.
Comment 5: p7, l187-p8, l212: I don't see a significantly higher tolerance towards glucose and ethanol when comparing the free and the immobilized ones. They are rather in a comparable range. Towards glucose the tolerance of the immobilized enzyme even seems to be a bit lower. If you insist on using the word significant, please perform proper statistical test on the data presented in figure 1e+f.
Reply 5: Thanks for the comment. The sentence has been rewritten to give greater clarity.
Comment 6: Please give for all figures using relative activities information on what 100% activity is in U/g in the legend. Makes it easier to compare among the different figures.
Reply 6: We include this information in the manuscript figure 1 and 2 legend.